# Observed Key Surface Parameters for Characterizing Land–Atmospheric Interactions in the Northern Marginal Zone of the Taklimakan Desert, China

**Lili Jin [1], Zhenjie Li [2], Qing He [1],*, Yongqiang Liu [3,4], Ali Mamtimin [1], Xinchun Liu [1], Wen Huo [1], Yu Xin [1], Jiantao Zhang [1] and Chenglong Zhou [1]**

1 Taklimakan Desert Meteorology Field Experiment Station of CMA, Institute of Desert Meteorology, China Meteorological Administration, Urumqi 830002, China; jinll@idm.cn (L.J.); ali@idm.cn (A.M.); liuxch@idm.cn (X.L.); huowenpet@idm.cn (W.H.); xiny@idm.cn (Y.X.); zhangjt@idm.cn (J.Z.); zhoucl@idm.cn (C.Z.)
2 Lincang Meteorological Bureau of Yunnan Province, Lincang 677099, China; lizhj@idm.cn
3 College of Resources & Environmental Science, Xinjiang University, Urumqi 830046, China; liuyq@idm.cn
4 Key Laboratory of Oasis Ecology (Xinjiang University) Ministry of Education, Urumqi 830046, China
* Correspondence: qinghe@idm.cn

**Abstract:** An observational data set of the year 2010 at a site in the northern marginal zone of the Taklimakan Desert (TD) was used to analyse the key surface parameters in land–atmospheric interactions in the desert climate of northwest China. We found that the surface albedo ($\alpha$) and emissivity ($\varepsilon$) were 0.27 and 0.91, respectively, which were consistent with the values obtained based on observations in the hinterland of the TD as well as being similar to the dry parts of the Great Basin desert in North America, where they were comparable to the $\alpha$ and $\varepsilon$ values retrieved from remote sensing products. Peak frequency value of $z_{0m}$ was $5.858 \times 10^{-3}$ m, which was similar to the Mojave Desert, Peruvian desert, Sonoran Desert, HEIFE (Heihe region) Desert, and Badain Jaran Desert. The peak frequency value of $z_{0h}$ was $1.965 \times 10^{-4}$ m, which was different from those obtained in the hinterland of the TD. The average annual value of excess resistance to heat transfer ($kB^{-1}$) was 2.5, which was different from those obtained in the HEIFE Gobi and desert, but they were similar to those determined for the Qinghai–Tibetan Plateau and HAPEX-Sahel. Both $z_{0m}$ and $z_{0h}$ varied less diurnally but notably seasonally, and $kB^{-1}$ exhibited weak diurnal and seasonal variations. We also found that $z_{0m}$ was strongly influenced by the local wind direction. There were many undulating sand dunes in the prevailing wind and opposite to the prevailing wind, which were consistent with the directions of the peak $z_{0m}$ value. The mean values calculated over 24 h for $C_d$ and $C_h$ were $6.34 \times 10^{-3}$ and $5.96 \times 10^{-3}$, respectively, which were larger than in the Gobi area, hinterland of the TD and semiarid areas, but similar to HEIFE desert. Under the normal prevailing (NNE–ESE) wind, the mean bulk transfer coefficient $C_d$ and $C_h$ were of the same order of magnitude as expected based on similarity theory. Using the data obtained under different wind directions, we determined the relationships between $C_d$, $C_h$, the wind speed $U$, and stability parameter $z/L$, and the results were different. $C_d$ and $C_h$ decreased rapidly as the wind speed dropped below 3.0 m s$^{-1}$ and their minimum values reached around 1–2 m s$^{-1}$. It should also be noted that the $\varepsilon$ values estimated using the sensible heat flux ($H$) were better compared with those produced using other estimation methods.

**Keywords:** eddy covariance technique; surface characteristic parameter; Taklimakan Desert

## 1. Introduction

Interactions between the atmosphere and the underlying surface play key roles in the land-air exchange of mass and energy for weather and climate changes [1,2]. In particular, many studies have observed and modeled land–atmospheric interactions in semiarid and arid regions because their natural environments and human populations are especially vulnerable to anomalous weather and climate conditions.

The physical processes of surface mass, energy and water exchange in the atmosphere are affected by several key parameters, such as the surface albedo ($\alpha$), surface emissivity ($\varepsilon$), aerodynamic and thermal roughness lengths ($z_{0m}$ and $z_{0h}$), and momentum and sensible heat drag coefficients ($C_d$ and $C_h$) [3–6]. Therefore, it is very important to study the variations in these parameters under different climate conditions. Thus, Boussetta et al. [7] showed that weather forecasts are sensitive to the surface albedo because it controls the partition of surface energy fluxes. In addition, $\varepsilon$ is a critical variable for calculating surface longwave radiation and it is used in both land and atmospheric remote sensing. Studies have shown that the surface temperature, outgoing longwave radiation, cooling rates, and frozen surface extent are sensitive to far-infrared surface emissivity [8,9]. $z_{0m}$ and $z_{0h}$ are defined as the height at which the wind speed becomes zero and at which the extrapolated air temperature is identical to the surface temperature over a homogeneous surface under neutral and thermally stratified conditions [10–12], respectively, and they are very important parameters for estimating the momentum, heat, and mass exchange between the surface and atmosphere [13–19]. They are generally estimated directly using the eddy covariance method and satellite data inversion technique [20,21]. Many previous studies have shown that $z_{0h}$ and $kB^{-1}$ exhibit diurnal and monthly variations [22–26]. In surface energy balance calculations, $C_d$ and $C_h$ are usually obtained from the bulk transfer equations [4]. Therefore, accurate estimates of these key parameters are important for land–air interaction observation experiments. However, we still lack a thorough understanding of the key parameters for representing land–atmospheric interactions and their underlying processes, thereby limiting our capacity for modeling and forecasting in desert.

In deserts, low vegetation density, soil surface organic carbon contents, and soil moisture lead to high $\alpha$ values (With more carbon and more soil moisture, the surface be darker, reducing albedo), which lower the net radiation compared with other ecosystems and with the same input of incoming solar radiation [27,28]. In addition, the surface energy residual exhibits strong diurnal variations due to the dominance of sensible heat. The $\alpha$ values also exhibit high seasonal variations due to frequent sand events during the sandstorm seasons [29–34]. The thermal effects of land surface processes play important roles in the formation of sandstorm weather, and dust aerosols in the atmospheric boundary layer can then have significant thermal dynamic and dynamic effects on the atmosphere [35]. The TD is an arid region of China and the second largest shifting sand desert on Earth; it has been studied widely based on variations in the characteristic parameters of land–atmospheric interactions and parameterization schemes [8,36–38], although we have limited knowledge of the spatial variations in the key parameters that dominate land–air coupling across this giant desert.

The TD (total area = ca 338,000 km$^2$) is located in the Tarim Basin, west China (Figure 1). It is the largest desert in China and it comprises 85% shifting sand with most types of sand dunes [39–41]. The TD strongly influences the climate and environment in the northwest arid region of China. Dust storms in this region also alter the energy balance and atmospheric chemistry [42,43]. In a recent study, Liu et al. [8,44] estimated the values of $\alpha$, $\varepsilon$, $z_{0m}$, and $z_{0h}$ in the hinterland of the TD. However, the information on key surface parameters in the northern marginal zone of the TD remains limited including $\alpha$, $\varepsilon$, $z_{0m}$, $z_{0h}$, $C_d$, and $C_h$. Thus, in this study, we aimed to analyze the key surface parameters ($\alpha$, $\varepsilon$, $z_{0m}$, $z_{0h}$, $kB^{-1}$, $C_d$, and $C_h$) at Xiaotang experimental field observation station in the northern marginal zone of the TD. In contrast to Tazhong (with an underlying surface of fine sand) where extensive observations have been made [8] and the site has slight distortions due to surrounding vegetation (*Gongliu*), the station (the underlying surface is an ancient river bed with a crust of clay and sand) is affected more evidently by nearby vegetation (*Populus diversifolia*).

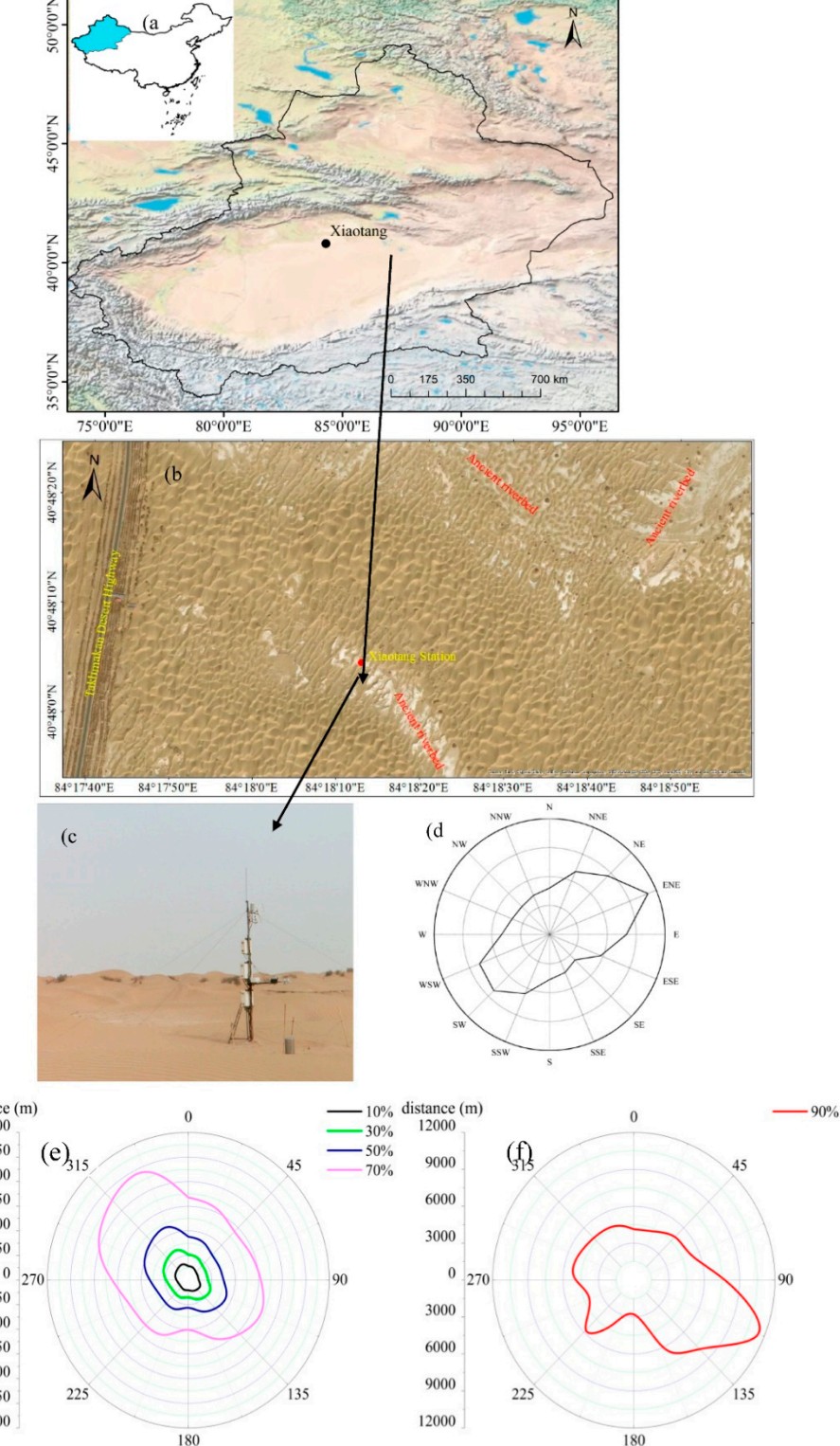

**Figure 1.** (**a**) Map of Xiaotang station in China and Xinjiang Province. (**b**) High resolution map of Xiaotang station. (**c**) Photograph of Xiaotang station in the Taklimakan Desert. (**d**) Wind direction at Xiaotang station during 5 January to 31 December 2010. (**e**) Flux footprint (10%, 30%, 50%, and 70%) at the observation station (concentric center), where the labels around the circle show the direction in geographical coordinates (0–360°) and the radius length of the circle is the distance. (**f**) Same as (**e**) but for 90%.

The important parameters ($\alpha$, $\varepsilon$, $z_{0m}$, $z_{0h}$, $C_d$, and $C_h$) were analyzed in a spatially inhomogeneous environment and compared with data obtained for other deserts or similar ecosystems. The remainder of this paper is organized as follows. In Section 2, we describe the observational site, instruments and data collection processes, quality control procedures, and methods used for deriving the key surface parameters. In Section 3, we present the observational analysis and comparisons with results obtained in Tazhong. Finally, we give our conclusions based on this analysis in Section 4.

## 2. Observational Site and Analytical Methods

### 2.1. Site Description

Observations obtained from the Xiaotang Land–Atmosphere Interaction Observation Station (933 m a.s.l., 40°48′ N, 84°18′ E) were used in this study. This station is located in the transitional zone (from oasis to desert ecosystem) over the northern margin of TD (Figure 1a–c). The station is in a sandy area that comprises sand and ancient river beds (Figure 1b,c).

According to observations obtained from Xiaotang meteorological station in 2010, the annual average temperature was 12.0 °C, and the average temperatures were 28.6 °C in July and −16.3 °C in January. This area is dominated by an extremely dry climate where the annual average precipitation is only 86.5 mm but the potential evaporation is 3025.2 mm. The prevailing wind direction is ENE (Figure 1d).

### 2.2. Data

The measurement levels, instruments, near-surface meteorological variables, turbulent fluxes, and surface radiation spectrum at the station are shown in Table 1.

**Table 1.** Measurement levels and instruments employed for acquiring near-surface meteorological variables and turbulent fluxes at Xiaotang station.

| Item | Height or Depth | Sensor |
|---|---|---|
| Wind direction, wind speed | 10 m | Metone 010C/020C |
| Solar radiation | 1.5 m | Kipp & Zonen CNR-1 (ventilated) |
| Longwave radiation | 1.5 m | Kipp & Zonen CNR-1 (ventilated) |
| Air temperature and humidity | 0.5, 1, 2, 4, 10 m | Vaisala HMP45D (ventilated) |
| Soil temperature | surface, 10, 20, 40 cm | Campbell 109L |
| Soil moisture | 2.5, 10, 20, 40 cm | Campbell CS616 |
| Soil heat flux | 2.5 cm, 8 cm | Hukseflux HFP01 |
| Turbulent fluxes | 3 m | Campbell CSAT3/Licor 7500 |
| Surface radiation spectrum of 8–14 μm | 1 m | Fourier transform infrared spectrometer (FTIR) |

The observations of meteorological variables and turbulent fluxes used in this study covered the period from 5 January to 31 December 2010. The observations of wind speed and direction at 30-min intervals were acquired by a Metone 010C/020C sensor at a height of 10 m and using a CR1000 data logger. Observations of radiation at 30-min intervals were obtained used a CRN-1 mounted at a height of 1.5 m and with a CR1000 data logger. Observations of the air temperature at 30-min intervals were made using a Vaisala HMP45D sensor at heights of 0.5, 1, 2, 4, and 10 m, and with a CR1000 data logger. Observations of the soil temperature were acquired at 30-min intervals using Campbell 109L sensors at the surface soil (half of the sensor was above the surface) at heights of 10, 20, and 40 cm above the surface, and with a CR1000 data logger. Raw turbulent flux data were obtained using a Campbell CSAT3/Licor 7500 sensor at a height of 3.0 m and with a CR5000 data logger.

The raw turbulent data were processed to obtain 30-min interval turbulent flux averages using EddyPro (Eddy Covariance Processing Software) software (the parameters employed in EddyPro are shown in Table 2). In addition, a series of quality control procedures were applied to the turbulent flux

data, i.e., noise removal, coordinate rotation, stability testing, variance similarity analysis, and removal of out-of-correct-range data. The statistical analyses showed that the proportions of missing data and unqualified data (abnormally high or low data) were 8.2% and 28.5%, respectively. The unqualified data were excluded.

**Table 2.** EddyPro parameters used in this study.

| Parameter | Value | Parameter | Value |
|---|---|---|---|
| File duration | 30 min | Acquisition frequency | 20 Hz |
| Canopy height | 0.1 m | Displacement height | 0 m |
| Roughness height | 0 m | Altitude | 933 m |
| Latitude | 40°48′ | Longitude | 84°18′ |
| Manufacturer | Campbell Scientific LI-COR | Model | CSAT-3 Li-7500 |
| Height | 3 m | Wind data format | $u, v, w$ |
| North offset | 0.0 [°] | Rotation method | Double rotation |

The Fourier transform infrared spectrometer (FTIR) was produced by Designs and Prototypes Company (USA). The instrument comprised of one Michelson interferometer, one detector, one black body, one electric current converter, one block reflector gold board, and one built-in individual computer. The computer program was used for sampling, processing, and the storage of samples on an individual computer. The FTIR had a fast recording speed, high signal-to-noise ratio, good sensitivity, and very little stray radiation, where the light flux = 0.016 cm$^2$ sr, spectral range = 2–16 μm, and spectral resolution = 2–24 cm$^{-1}$. The measurement results obtained had standard deviations less than 1% [45].

The principle of FTIR involves converting the received infrared spectra $M_s(\lambda)$ into spectral power $Vs(\lambda)$ by using the internal photoelectric effect:

$$Vs(\lambda) = \left| r(\lambda)M_s(\lambda) + r(\lambda)M^0(\lambda, T_{inst}) \right| \tag{1}$$

where $\lambda$ is the wavelength, $r(\lambda)$ is the linear response of the FTIR, and $M^0(\lambda, T_{inst})$ is radiation at the temperature of the FTIR. The FTIR was calibrated before measuring the radiation from the sample. $M_s(\lambda)$ is the sample radiation measured by the FTIR after calibration.

According to Kirchhoff's law, for opaque objects such as surfaces, the sum of the absorptivity and reflectivity ($R_s$) is 1, and the absorptivity is equal to $\varepsilon$; thus,

$$R_s = 1 - \varepsilon_s(\lambda) \tag{2}$$

The distance from the ground measurement sample to the sensor was less than 1 m, where the upward radiation could be treated as negligible at this distance and the atmospheric transmittance was considered to be 1. Thus, the radiation from the sample was expressed as:

$$M_s(\lambda) = \varepsilon_s(\lambda)B(\lambda, T_s) + [1 - \varepsilon_s(\lambda)]M_{dw\tau}(\lambda), \tag{3}$$

where $\varepsilon_s(\lambda)B(\lambda, T_s)$ is the radiation at temperature $T_s$. $[1 - \varepsilon_s(\lambda)]M_{dw\tau}(\lambda)$ is the radiation that is reflected by environmental influences such as the atmosphere. $M_{dw\tau}(\lambda)$ is the fallout radiation comprising downward radiation in the atmosphere reflected by the sample detected with the FTIR. $B(\lambda, T_s)$ is black body radiation.

$\varepsilon_s(\lambda)$ was obtained by transforming formula (3) [45,46], as follows,

$$\varepsilon_s(\lambda) = \frac{M_s(\lambda) - M_{dw\tau}(\lambda)}{B(\lambda, T_s) - M_{dw\tau}(\lambda)} \tag{4}$$

In Equation (4), $M_s(\lambda)$ and $M_{dw\tau}(\lambda)$ were obtained by the FTIR. Thus, sample calibration was necessary before measuring $M_s(\lambda)$ and $M_{dw\tau}(\lambda)$ to ensure the accuracy of the measurements.

Therefore, $\varepsilon_s(\lambda)$ was converted into $\varepsilon$:

$$\varepsilon = \frac{\int_{\lambda_1}^{\lambda_2} \varepsilon_s(\lambda) B(\lambda, T_s) d\lambda}{\int_{\lambda_1}^{\lambda_2} B(\lambda, T_s) d\lambda} \tag{5}$$

where $\lambda_1$ and $\lambda_2$ are wavelength ranges with thermal infrared atmospheric window values of 8 and 14, respectively. In order to facilitate the computation, the integral equation was discretized as follows.

$$\varepsilon = \frac{\sum_{\lambda=\lambda_1}^{\lambda_2} \varepsilon_s(\lambda) B(\lambda, T_s) \Delta\lambda}{\sum_{\lambda=\lambda_1}^{\lambda_2} B(\lambda, T_s) \Delta\lambda} \tag{6}$$

In fact, in order to improve the accuracy, the wavelength range interval of 8–14 μm was divided into 375 $\Delta\lambda$.

Land use is the same in all directions in TD (large scale), but the sand dunes are large and undulating near to the station. There is a flat and bare ancient riverbed SE (NW) of the station's tower, which was consistent with the directions of 90% (70%) of the flux source area. The flux source area could be observed well in the windward direction (prevailing wind direction) [47]. In this study, the bulk transfer coefficients ($C_d$ and $C_h$) were calculated based only on data measured in the NNE–ESE wind directions (local dominant wind direction) (Figure 1d–f). We then compared the results with those obtained in all wind directions.

The ground surface soil heat flux was calculated according to the soil temperature and moisture gradients [48]. Due to errors in the land surface temperature during the sand dust season, the ground surface temperature was calculated using the observed $\varepsilon$ and radiation values.

### 2.3. Theory and Methodology

According to many other observational studies [49,50], the energy balance closure is not achieved completely for different underlying types. The energy balance closure is usually formulated as: $R_n - G_0 = H + LE$, where $R_n$ is the net radiation flux, $G_0$ is the ground surface soil heat flux, $H$ is the sensible heat flux, and $LE$ is the latent heat flux. The ratio of the energy balance closure is usually formulated as: $EBR = \sum(H + LE)/\sum(R_n - G_0) * 100\%$. The energy balance is formulated relative to the residuals as: $\delta = [(R_n - G_0) - (H + LE)]/(R_n - G_0)$. $H + LE < R_n - G_0$ when $\delta > 0$, while $H + LE > R_n - G_0$ when $\delta < 0$.

In the analysis, $\alpha$ was calculated from the observed surface solar radiation components [51] using Equations (7) and (8):

$$\alpha = S_\uparrow / S_\downarrow \tag{7}$$

$$\overline{\alpha} = \sum_{i=1}^{n} \frac{S_{\uparrow i}}{\sum_{i=1}^{n} S_{\uparrow i}} \alpha_i \tag{8}$$

where $S_\uparrow$ is shortwave upward radiation, $S_\downarrow$ is shortwave downward radiation, $\overline{\alpha}$ is the mean of $\alpha$ (calculated using the weighted mean method), and the subscript $i$ is the time index.

In this study, we focused on the measurements in sunny and dry weather. In order to calibrate the radiation, the FTIR had to be calibrated every 10–20 min to a black body. The temperature of the cold black body was 10 °C lower than the environmental temperature, whereas the temperature of the hot black body was 10 °C higher than the surface temperature. After setting the temperatures of the cold black body and the hot black body, radiation spectrum data were measured for the cold and hot black body and saved. The accuracy achieved for the black body emissivity was 0.994–0.998 ± 0.002 and the accuracy of the temperature was ±0.1 °C. Thus, the error caused by the black body was less

than 0.004. The temperature fluctuations according to the interferometer remained within 0.1 °C and the calibration error for the black body was less than 0.002 [46]. In order to reduce the interference in the instrument's noise signal, we set the spectrum stacking number at 10 times and the average values were obtained.

In this study, we performed three steps to minimize the error during operation in order to obtain the surface emissivity spectrum with high accuracy: (1) we measured the cold black body, hot black body, and diffuse reflection radiation from a gold plate; (2) we measured the surface radiation; and (3) we repeated step (1). These three steps were performed as quickly as possible, where we limited the time to 10 min for each group of measurements. Radiation correction was performed in steps (1) and (3) to evaluate the influence of the environment over time on the surface spectral radiation, thereby reducing the error. The surface temperature of the diffuse gold plate was measured using a thermoelectric coupling thermometer. In general, the average value was taken based on five measurements.

For the desert surface, Korb et al. [45] suggested that the maximum emission rate for the band fitted (black body radiation spectrum fitted to the surface radiation spectrum) at 7.45–7.65 µm is 0.95. By using this method to obtain the surface radiation temperature, the surface emission spectra were acquired for a thermal infrared window at wavelengths of 8–14 µm in the TD with high efficiency and accuracy.

We used the physically-based and semi-empirical methods of Yang et al. [11] to calculate $\varepsilon$. The computed surface sensible heat ($H_{sfc}$) was compared with the observed surface sensible heat ($H_{obs}$) to fit the $\varepsilon$ values. $H_{sfc}$ can be obtained from: $H_{sfc} = \rho C_p (T_0 - T_a)/r_h$, where $C_p$ [=1004 J kg$^{-1}$ K$^{-1}$] is the heat capacity of air at constant pressure, $\rho$ is the air density (kg m$^{-3}$), $T_0$ is the aerodynamic surface temperature (K), $T_a$ is the air temperature (K), and $r_h$ is the aerodynamic resistance for heat (s m$^{-1}$) $(r_h = \Pr\left(\ln\frac{z_m}{z_{0m}} - \psi_m(\zeta)\right)\left(\ln\frac{z_h}{z_{0h}} - \psi_h(\zeta)\right)(k^2 U)^{-1})$. $T_0 = \left[\frac{R_{lw}^{\uparrow} - (1-\varepsilon)R_{lw}^{\downarrow}}{\varepsilon\sigma}\right]^{1/4}$, where Pr is the Prandtl number (=1 if $z/L \geq 0$ and 0.95 if $z/L < 0$), $L[\equiv T_a u_*^2/(kgT_*)]$ is the Obukhov length, $u_*$ is the frictional velocity (m s$^{-1}$); $T_*[\equiv -H/(\rho c_p u_*)]$ is the frictional temperature; $\psi_m(\zeta)$ and $\psi_h(\zeta)$ are the integrated stability correction function for momentum transfer and temperature profiles, respectively; $k$ (=0.4) is the von Kármán constant, $R_{lw}^{\uparrow}$ and $R_{lw}^{\downarrow}$ are the upward longwave radiation and downward longwave radiation, respectively. $\sigma$ (5.677 × 10$^{-8}$ W m$^{-2}$ K$^{-4}$) is the Stefan–Boltzmann constant near neutral (i.e., $T_0 = T_a$). The difference between $H_{sfc}$ and $H_{obs}$ is sensitive to the value of $\varepsilon$. Using an iterative algorithm ($\varepsilon$ values from 0.8 to 1.0 with a step width of 0.001), the $\varepsilon$ value was derived by minimizing the root mean square (RMS) between the calculated $H_{sfc}$ and observed $H_{obs}$.

In the analysis, $z_{0m}$ was estimated using Equations (9) to (11):

$$\ln z_{0m} = \ln(z - d) - \frac{ku}{u_*} - \psi_m(\zeta), \quad \zeta = (z - d)/L \tag{9}$$

where $z$ is the observed height (m) and $d$ is the displacement height (m), which was negligible in our study with no vegetation coverage.

According to Dyer ([52], parameters 16 and 5 in Equations (10) and (11) are consistent with $k = 0.40$, while $u$ is the average wind velocity (m s$^{-1}$). Under unstable conditions [12,53], the following equation can be used:

$$\psi_m(\zeta) = 2\ln(\frac{1+x}{2}) + \ln(\frac{1+x^2}{2}) - 2\arctan x + \frac{\pi}{2}, \quad \zeta < 0, \tag{10}$$

where, $x = (1 - 15.2\zeta)^{1/4}$.

Under stable conditions [54,55], the equation is as follows.

$$\psi_m(\zeta) = -4.8\zeta, \quad \zeta > 0 \tag{11}$$

The value of $z_{0h}$ is difficult to determine because it cannot be measured directly, but it is possible to derive $z_{0h}$ from the equations for $H_{obs}$ [23], where it can be obtained from the flux-gradient relationships in a surface layer based on Monin–Obukhov (M–O) similarity theory [56]. The following equation can be used.

$$\ln z_{0h} = \ln(z - d) - \frac{ku_*(T_0 - T_a)}{H_{obs}/(\rho C_p)} - \psi_h(\zeta) \tag{12}$$

Under unstable conditions:

$$\psi_h(\zeta) = 2\ln(\frac{1 + x^2}{2}), \quad \zeta < 0 \tag{13}$$

Under stable conditions:

$$\psi_h(\zeta) = -5\zeta, \quad \zeta > 0 \tag{14}$$

where $x = (1 - 16\zeta)^{1/4}$ and $\zeta = (z - d)/L$.

In the analysis, we employed the same procedure used by Yang et al. [11] for data quality control, as follows: (1) excluding periods when the absolute value of $|H_{obs}| < 10$ W m$^{-2}$, which corresponds to the time with a low solar elevation angle when the observation error was large; (2) excluding periods when $z_{0h} > z_{0m}$, which is physically unrealistic; and (3) excluding data from sandy, rainy, and snowy periods for $C_d$ and $C_h$.

In our analysis, $C_d$ and $C_h$ were calculated using two methods. First, we followed the Eddy correlation method using Equation (15):

$$\begin{cases} C_d = \dfrac{\tau}{\rho u^2}, \\ C_h = \dfrac{H}{\rho c_p u(T_0 - T_a)} \end{cases} \tag{15}$$

where $\tau$ is the surface stress (kg m$^{-1}$ s$^{-2}$).

The other method for calculating $C_d$ and $C_h$ is based on M–O similarity theory where $C_d$ and $C_h$ are expressed as [57,58]:

$$\begin{cases} C_d = \dfrac{k^2}{[\ln((z - d)/z_{0m}) - \psi_m((z - d)/L)]^2}, \\ C_h = \dfrac{k^2}{[\ln((z - d)/z_{0m}) - \psi_m((z - d)/L)][\ln((z - d)/z_{0h}) - \psi_h((z - d)/L)]} \end{cases} \tag{16}$$

where $\Psi_m(z/L)$ and $\Psi_h(z/L)$ are the integration forms of the M–O similarity functions $\varphi_m$ and $\varphi_h$ at the station, respectively, and the equation is:

$$\begin{cases} \phi_m = \dfrac{k}{u_*} \dfrac{\Delta u}{\ln(z_2/z_1)} \\ \phi_h = \dfrac{k}{\theta_*} \dfrac{\Delta \theta}{\ln(z_2/z_1)} \end{cases} \tag{17}$$

where $\Delta$ is the difference symbol, $z_1$ and $z_2$ are the height above ground (in this analysis, $z_1 = 2$ m, mboxemphz$_2 = 4$ m), $\theta_*$ is the turbulent temperature scale, and $\theta$ is the potential temperature.

The lack of observations across the whole TD prevented us from capturing the spatial distributions of these surface parameters based on in situ observations alone, so we also compared our site observations with the values of $\alpha$ and $\varepsilon$ from remote sensing products. This comparison allowed us to use in situ observations to calibrate the remote sensing products and to produce improved surface parameter estimates over a large spatial domain. In this analysis, Landsat 8 data from NASA were used to compare the observation data for $\alpha$ and $\varepsilon$. The empirical method described by Qin et al. [59]

was used to calculate $\alpha$ and $\varepsilon$ from the Landsat data. We only used data from 19 August 2013 at 11:00 (Local time), which was a sunny day. The solar elevation angle at the station was $59.05°$ at 11:00 on 19 August 2013.

## 3. Results and Discussion

### 3.1. Surface Energy Balance Closure

The energy balance closure was not achieved completely at the station (Figure 2a). During the daytime (positive net radiation), *EBR* was about 40–60%, whereas it was less than 25% at night (negative net radiation) with no correction for the soil heat flux (Figure 2a). Figure 2b shows that during the daytime, *EBR* was about 60–70%, whereas it was 25–35% at night with $G_0$. Table 3 shows during the dust season (March and April), *EBR* was less than 60%, whereas it was more than 60% during other seasons (it was not possible to obtain statistics for *EBR* due to the energy balance data synchronization matching problem in October, November, December, January, and February).

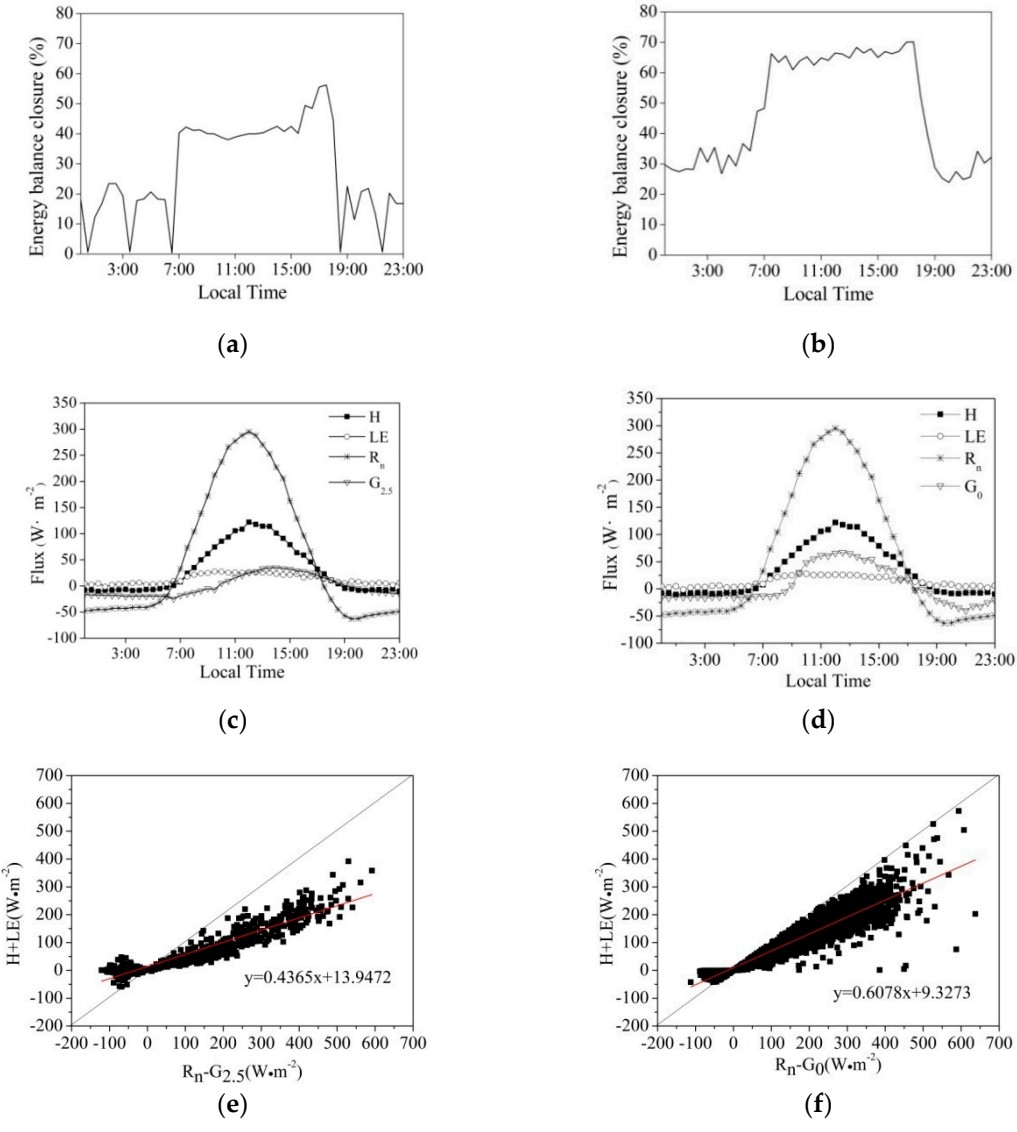

**Figure 2.** *Cont.*

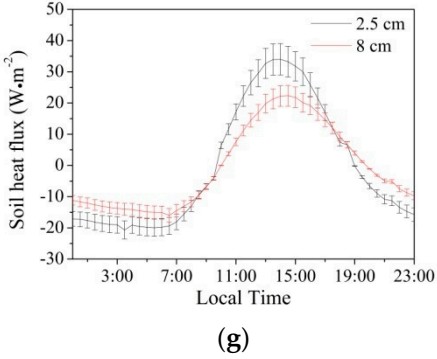
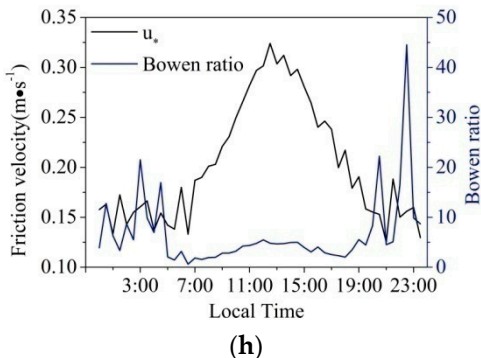

**(g)** **(h)**

**Figure 2.** (**a**) Mean daily variations in the energy closure from 5 January to 31 December 2010. (**b**) Same as (**a**) but corrected for the soil heat flux. (**c**) Mean daily variations in the flux (*H* is the surface sensible heat flux, *LE* is the latent heat flux, $R_n$ is the net radiation, and $G_0$ is the surface soil heat flux). (**d**) Same as (**c**) but corrected for the soil heat flux. (**e**) Scatter plot with regression for $H + LE$ and $R_n - G_{2.5}$. (**f**) Scatter plot with regression for $H + LE$ and $R_n - G_0$. (**g**) Mean daily variations in the soil heat flux at depths of 2.5 cm and 8 cm. (**h**) Mean daily variations in the friction velocity and Bowen ratio.

**Table 3.** *EBR* in every month.

| Month | March | April | May | June | July | August | September |
|--------|-------|-------|------|------|------|--------|-----------|
| *EBR* | 53.4% | 57.8% | 68.5% | 68.3% | 71.2% | 66.4% | 71.3% |
| Sample | 636 | 510 | 337 | 982 | 1346 | 776 | 71 |

Figure 2c shows the mean diurnal cycles of the energy fluxes. The monthly mean peak *H*, *LE*, $R_n$, and 2.5-cm depth soil heat flux ($G_{2.5}$) reached 122.1, 27.7, 295.1, and 35.1 W m$^{-2}$, respectively; *H* and $R_n$ exhibited different patterns compared with LE and $G_{2.5}$. Figure 2d,g shows that the monthly mean peak $G_0$ and 8-cm depth soil heat flux ($G_8$) reached 64.2 W m$^{-2}$ and 22.3 W m$^{-2}$, respectively. The slope value of $H + LE$ and $R_n - G_{2.5}$ was around 0.44, and that of $H + LE$ and $R_n - G_0$ was around 0.61. In the regressions, $G_0$ increased the slope of the ordinary least squares regression by 39%, which is why it was necessary to determine the ground heat storage in the desert in this study. Over the desert, LE was a very small component of the surface energy balance with the weak diurnal variation where the maximum values for LE and $G_0$ ($G_{2.5}$ and $G_8$) occurred in the morning and noon (afternoon), respectively, because the soil heat capacity was larger than the air heat capacity. *EBR* was typically lower when the friction velocity was below 0.19 m s$^{-1}$, but greater when the friction velocity was above the 0.19 m s$^{-1}$ (Figure 2h). *EBR* was typically lower when the Bowen ratio was above 4.6, but greater when the Bowen ratio was below the 4.6 (Figure 2h). Wilson et al. [49] suggested that the main reasons for the energy imbalance are systematic errors associated with sampling mismatch, systematic instrument bias, neglected energy sinks, low and high frequency loss of turbulent fluxes, and horizontal and/or vertical advection of heat and water vapor. Previous studies have suggested that the energy balance is almost closed in a flat desert region. However, the energy imbalance is large in the TD. At the station, the neglected surface soil heat flux was corrected based on the soil temperature and humidity; thus, it was mainly due to the neglected low frequency or high frequency turbulent flux. In spite of this, landscape heterogeneity at the station cannot be ignored as a contributor to incomplete energy balance closure [50].

### 3.2. Relationships between $\varphi_m$, $\varphi_h$, and z/L

Figure 3 shows the observed and calculated functions $\varphi_m$ and $\varphi_h$ against $z/L$ in the NNE–ESE wind direction. In this analysis, data validation showed that data were in agreement with the theoretical relationship between the functions $\varphi_m$, $\varphi_h$, and $z/L$ at the station under the same wind direction.

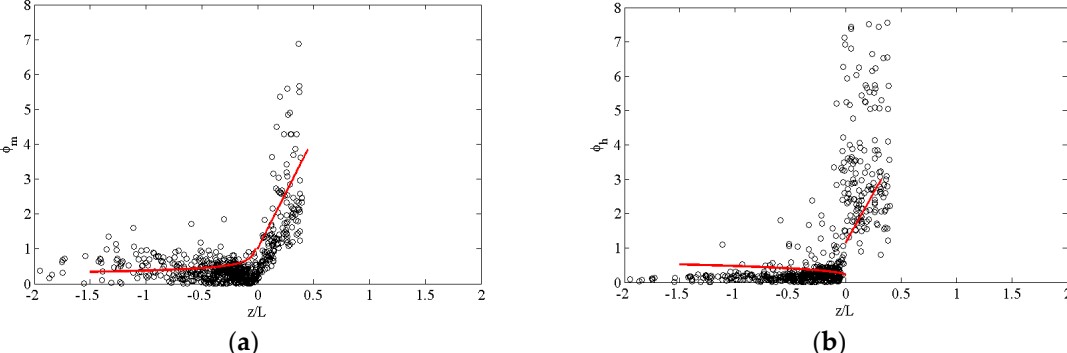

(**a**)                                          (**b**)

**Figure 3.** (**a**) Observed and calculated functions of $\varphi_m$ against $z/L$. (**b**) Same as (**a**) but for $\varphi_h$.

The relationships between $\varphi_m$, $\varphi_h$, and $z/L$ in the NNE–ESE wind direction were obtained as follows (Equations (18) and (19)), where these coefficients are based on the least square fits.

$$\begin{cases} \phi_m = 1 + 6.328\dfrac{z}{L}, \dfrac{z}{L} > 0, \\[2mm] \phi_h = 0.2279(1 - 17.27\tfrac{z}{L})^{\frac{1}{4}}, \dfrac{z}{L} \leq 0 \end{cases} \tag{18}$$

$$\begin{cases} \phi_m = 3.278(1 + 0.5387\dfrac{z}{L}), \dfrac{z}{L} > 0, \\[2mm] \phi_h = 3.515(1 - 7547\tfrac{z}{L})^{-\frac{1}{4}}, \dfrac{z}{L} \leq 0 \end{cases} \tag{19}$$

### 3.3. Surface Albedo α

To remove any possible contamination of clouds from $\alpha$ measurements, the data acquired during sunny periods were used for estimating this parameter. Figure 4a shows the relationship between $\alpha$ and the solar elevation angle ($h$) during sunny periods, which demonstrates that there was a significant exponential relationship between $\alpha$ and $h$ (when taking it as a value) where $\alpha$ was fitted best by the following numerical value equation.

$$\alpha = 0.2586 + 0.2415e^{-h/8.852} \tag{20}$$

Figure 4b shows the relationship between $\alpha$ and $h$ during cloudy periods by the following numerical value Equation (21).

$$\alpha = 0.4424h^{-0.1404} \tag{21}$$

According to Figure 4a,b, $\alpha$ was 0.27 when $h > 15°$, which is in good agreement with the values obtained for other deserts, as well as the measurements from Tazhong station at the center of the TD [3,36,40,60,61]. $\bar{\alpha}$ (weighted mean) was 0.31 and 0.30 on the sunny days and cloudy days, respectively. According to the regression-like analysis of the outgoing versus incoming shortwave radiation during sunny and cloudy periods (Figure 4c,d), the results were similar to the weighted mean values, but the outgoing shortwave radiation was slightly lower when the assumptions underlying the weighted mean were met. The diurnal variations in $\alpha$ during sunny and cloudy periods are shown in Figure 4e. $\alpha$ was higher in the morning and evening, but lower in the daytime, where the values was

0.27 from 9:00 to 15:00 (LST) during sunny and cloudy periods. $\alpha$ was slightly lower during cloudy periods than sunny periods, especially in the morning. We also compared the observed values with the values of $\alpha$ derived from the Landsat 8 satellite over this area. The retrieved albedos (from Landsat 8 data) were close to the observed values at both the station and Tazhong, where they ranged from 0.24 to 0.30 near the station (Figure 4f). Therefore, our results suggest that the satellite-derived values of $\alpha$ from remote sensing products are highly reliable for use in the TD.

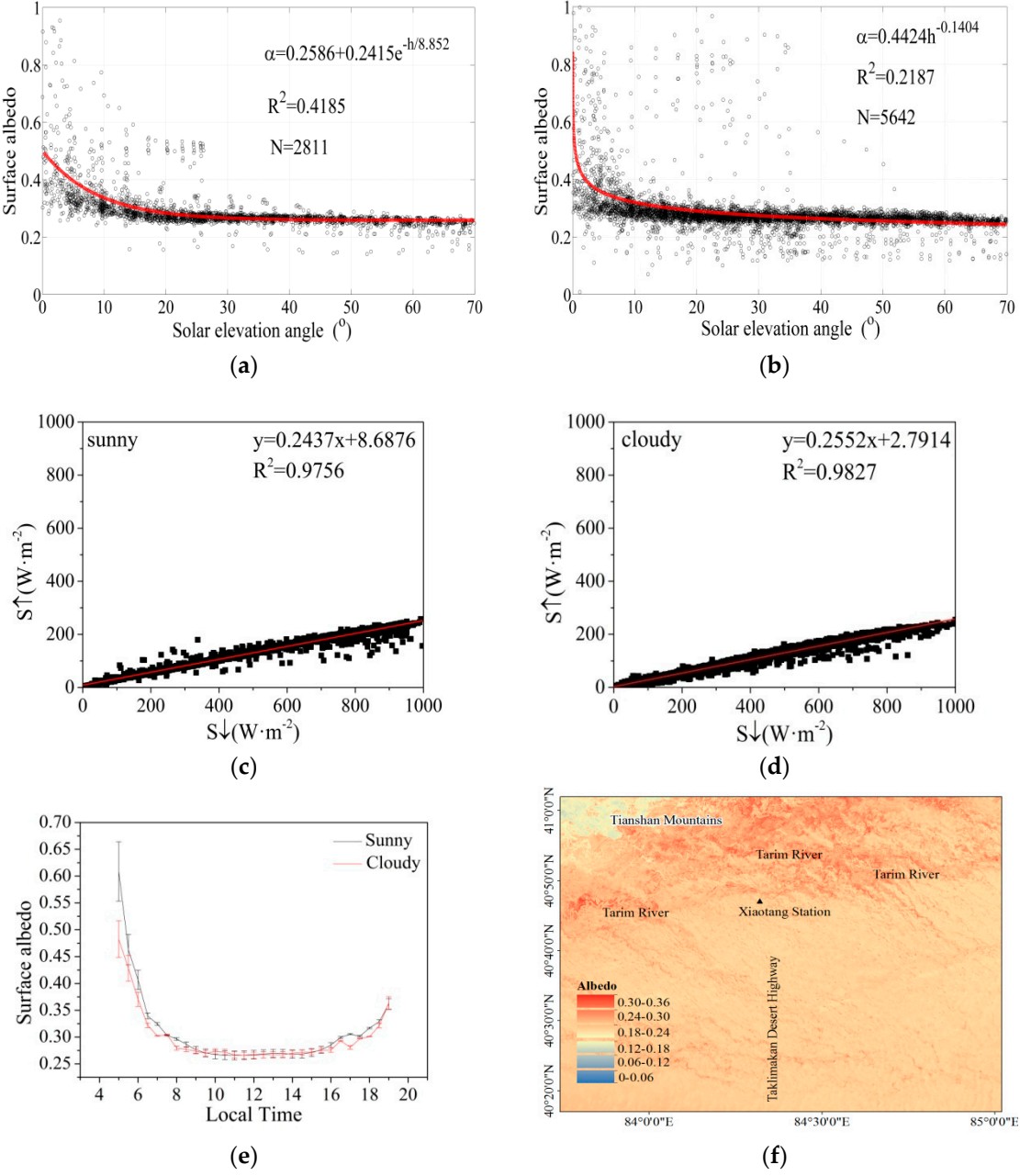

**Figure 4.** (**a**) Changes in the surface albedo with solar elevation angles on sunny days at Xiaotang station during 5 January to 31 December 2010. (**b**) Same as (**a**) but on cloudy days. (**c**) Scatter plot of outgoing against incoming radiation on sunny days. (**d**) Same as (**c**) but on cloudy days. (**e**) Diurnal variations in the surface albedo on sunny days and cloudy days. (**f**) Retrieval of the surface albedo from Landsat data.

### 3.4. Surface Emissivity ε

Figure 5a shows that the value of $\varepsilon$ corresponding to the minimum RMS error value for $H_{sfc}$ against $H_{obs}$ was 0.85 under near neutral conditions. Thus, we determined the sensitivity of $\varepsilon$ to $H_{obs}$. Table 4 shows the difference in $\varepsilon$ with $H_{obs}$ according to the method employed, which indicates that $\varepsilon$ decreased (increased) as $H_{obs}$ increased (decreased). For instance, $\varepsilon$ increased by 2, 7.6, 9, 11.6, 12.8, 13.5, and 17.4% when $H_{obs}$ increased by 5, 10, 15, 20, 25, 30, and 35%, respectively, whereas $\varepsilon$ decreased by 5.2, 8.9, 16.7, 23.4, 28.2, 33.3, and 37.6% when $H_{obs}$ decreased by 5, 10, 15, 20, 25, 30, and 35%, respectively.

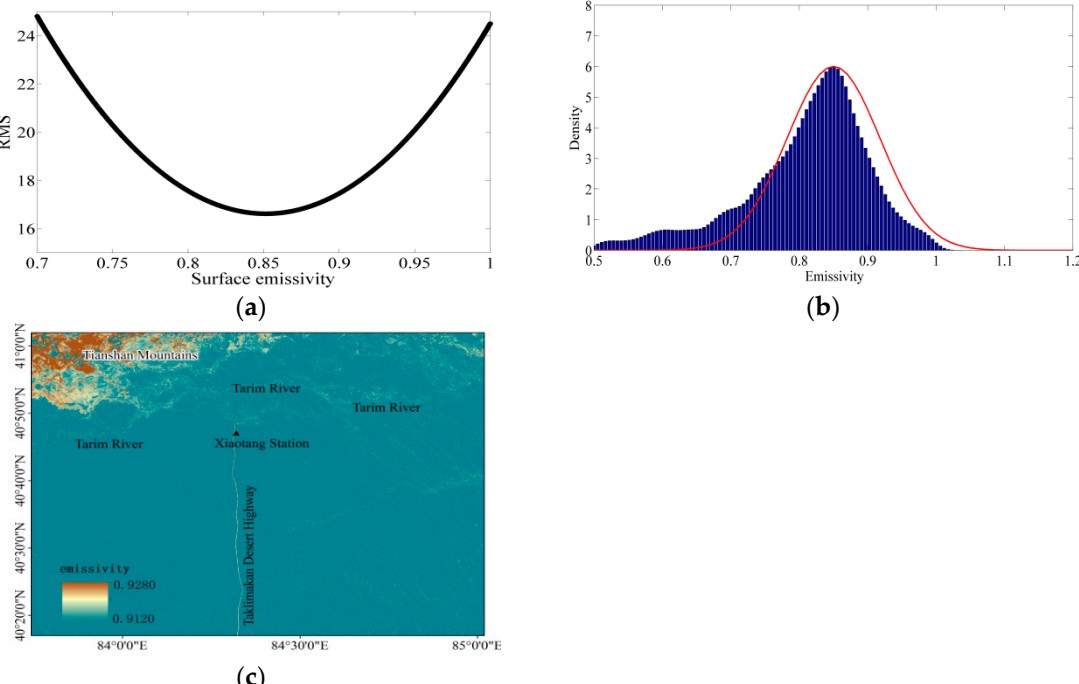

**Figure 5.** (**a**) Dependence of the surface emissivity ($\varepsilon$) on the RMS in $H_{obs}$ and $H_{sfc}$ in near neutral conditions at Xiaotang from 5 January to 31 December 2010. (**b**) Dependence of the surface emissivity ($\varepsilon$) on density according to the surface temperature method. (**c**) Retrieval of the surface emissivity from Landsat data.

**Table 4.** Difference in $\varepsilon$ with $H_{obs}$ according to the method employed.

| $H_{obs}$ Variation | $\varepsilon$ |
|---|---|
| −40% | invalid |
| −35% | 0.532 |
| −30% | 0.568 |
| −25% | 0.612 |
| −20% | 0.653 |
| −15% | 0.710 |
| −10% | 0.776 |
| −5% | 0.808 |
| default | 0.852 |
| +5% | 0.876 |
| +10% | 0.917 |
| +15% | 0.929 |
| +20% | 0.951 |
| +25% | 0.961 |
| +30% | 0.967 |
| +35% | 1 |

Figure 5b shows that the dependence of $\varepsilon$ on the density according to the surface temperature method was 0.85. Table 5 shows that the values of $\varepsilon$ averaged over a temperature difference of 0–0.1 °C in group 1 (0.839) and group 2 (0.818), estimated according to temperature gradients in near zero conditions, which was about 0.83, and the mean value of $\varepsilon$ averaged over a temperature difference of 0.1–0.2 °C in group 1 (0.849) and group 2 (0.825) estimated according to temperature gradients in near zero conditions, which was about 0.84, and so on. We also selected the same day to retrieve the value of $\varepsilon$ for these deserts using the Landsat 8 data.

**Table 5.** Estimates of $\varepsilon$ according to the temperature gradients in near zero conditions.

| Groups | Height (m) | Temperature Difference (°C) | Data Records | $\varepsilon$ |
|---|---|---|---|---|
| 1 | 0.5 | 0.5–1.0 | 71,437 | 0.798 |
|  | 1 | 0.4–0.5 | 2011 | 0.849 |
|  | 2 | 0.3–0.4 | 2556 | 0.849 |
|  | 4 | 0.2–0.3 | 3131 | 0.850 |
|  |  | 0.1–0.2 | 3961 | 0.849 |
|  |  | 0–0.1 | 3893 | 0.839 |
| 2 | 0.5 | 0.5–1.0 | 117,874 | 0.798 |
|  | 1 | 0.4–0.5 | 4929 | 0.854 |
|  | 2 | 0.3–0.4 | 6310 | 0.852 |
|  | 4 | 0.2–0.3 | 6272 | 0.834 |
|  | 10 | 0.1–0.2 | 6564 | 0.825 |
|  |  | 0–0.1 | 5857 | 0.818 |

Figure 5c shows that the retrieved $\varepsilon$ values (from Landsat 8 data) were generally close to the observed value, with a range of 0.912 to 0.916 at the station. The observed value of $\varepsilon$ was 0.91 according to the FTIR data, which is very close to that reported for other deserts, i.e., 0.91 to 0.97 [36,40,62]. However, $\varepsilon$ estimated based on sensible heat flux and surface temperature was both 0.85, which was lower than the observed value (0.91).

### 3.5. Aerodynamic Roughness Length ($z_{0m}$), Thermal Roughness Length ($z_{0h}$), and $kB^{-1}$

We used Equations (9)–(11) to estimate the aerodynamic roughness length $z_{0m}$ and Equations (12)–(14) to estimate the thermal roughness length $z_{0h}$.

Figure 6a,b shows that the distributions of $z_{0m}$ and $z_{0h}$ were approximately log-normal. The peak frequency for $\ln(z_{0m})$ was $-5.14$, which was equivalent to $z_{0m} = 5.858 \times 10^{-3}$ m in the normally distributed histogram. Similar values were obtained for deserts such as the Mojave Desert, Peruvian desert, Sonoran Desert, HEIFE (Heihe region) Desert, and Badain Jaran Desert, which were all in the order of $10^{-3}$ m [40,62–64]. Figure 6b shows that the peak frequency was $-8.54$ for $\ln(z_{0h})$, which was equivalent to $z_{0h} = 1.965 \times 10^{-4}$ m in the normally distributed histogram. Data with magnitudes of $10^{-3}$ ($z_{0m}$) and $10^{-4}$ ($z_{0h}$) were retained in the analysis.

Many previous studies have investigated the relationships between $z_{0m}$, the wind speed, the wind direction [12,65], and $u_*$ [66]. $z_{0m}$ does not change with the wind direction in areas with a flat and uniform underlying surface, whereas $z_{0m}$ might be related to the wind direction in areas with a non-uniform underlying surface (rough elements are not evenly distributed in different directions).

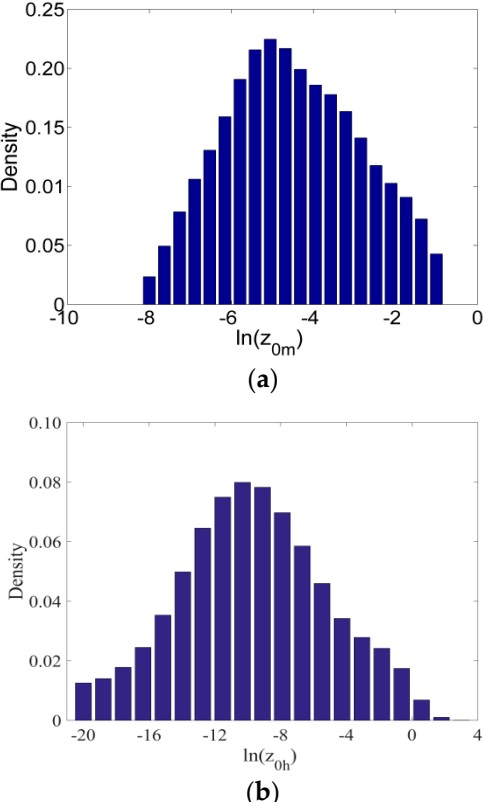

**Figure 6.** (**a**) Frequency distribution of the estimated $\ln(z_{0m})$ parametric density at Xiaotang station from 5 January to 31 December 2010. The optimal value of $\ln(z_{0m})$ was $-5.14$, which has the highest frequency in the curve. (**b**) Same as (**a**) but for $\ln(z_{0h})$ where $\ln(z_{0h})$ was $-8.54$.

Figure 7a,b shows the changes in $z_{0m}$ and $z_{0h}$ with different wind directions at the station. $z_{0m}$ had a double peak when the wind was in the E direction and the SW–WSW wind direction. $z_{0m}$ was $3.2 \times 10^{-2}$ m when the wind was in the E wind direction and $2.8 \times 10^{-2}$ to $3 \times 10^{-2}$ m when it was in the SW–WSW wind direction at the station. In addition, $z_{0h}$ had a double peak when the wind was in the ENE wind direction and the NNW wind direction. $z_{0h}$ was $5.3 \times 10^{-3}$ m when the wind was in the ENE wind direction and $5.6 \times 10^{-3}$ m when it was in the NNW wind direction at the station.

Figure 7c,d shows the changes in $z_{0m}$ and $z_{0h}$ with different wind directions at the station, where the data magnitudes were $10^{-3}$ and $10^{-4}$ for $z_{0m}$ and $z_{0h}$, respectively. $z_{0m}$ had two high values in areas when the wind was in the NE–E wind direction and the S–WSW wind direction. $z_{0m}$ was $5.6$–$6.3 \times 10^{-3}$ m when the wind was in the NE–E wind direction and $5.1$–$6.1 \times 10^{-3}$ m when it was in the S–WSW wind direction at the station. However, $z_{0h}$ only changed weakly with the wind direction. Figure 1d shows the characteristic rough surface elements at the station.

Therefore, the obvious changes in $z_{0m}$ with the wind direction at the station were affected mainly by the spatial inhomogeneity. The distributions of the surrounding sand dunes at the station are not uniform. In the NE–E and SSE–WSW directions from the tower (red dot in Figure 1), there are many undulating sand dunes, which were consistent with the directions of the peak $z_{0m}$ value. A flat and bare ancient riverbed is located in the SE direction from the tower, which is consistent with the directions of the 90% flux source area and the small values of $z_{0m}$. Thus, the results were roughly consistent with the fact that the peak $z_{0m}$ values occurred with the prevailing wind and opposite to the prevailing wind, whereas they did not occur in the 90% flux source area at the station.

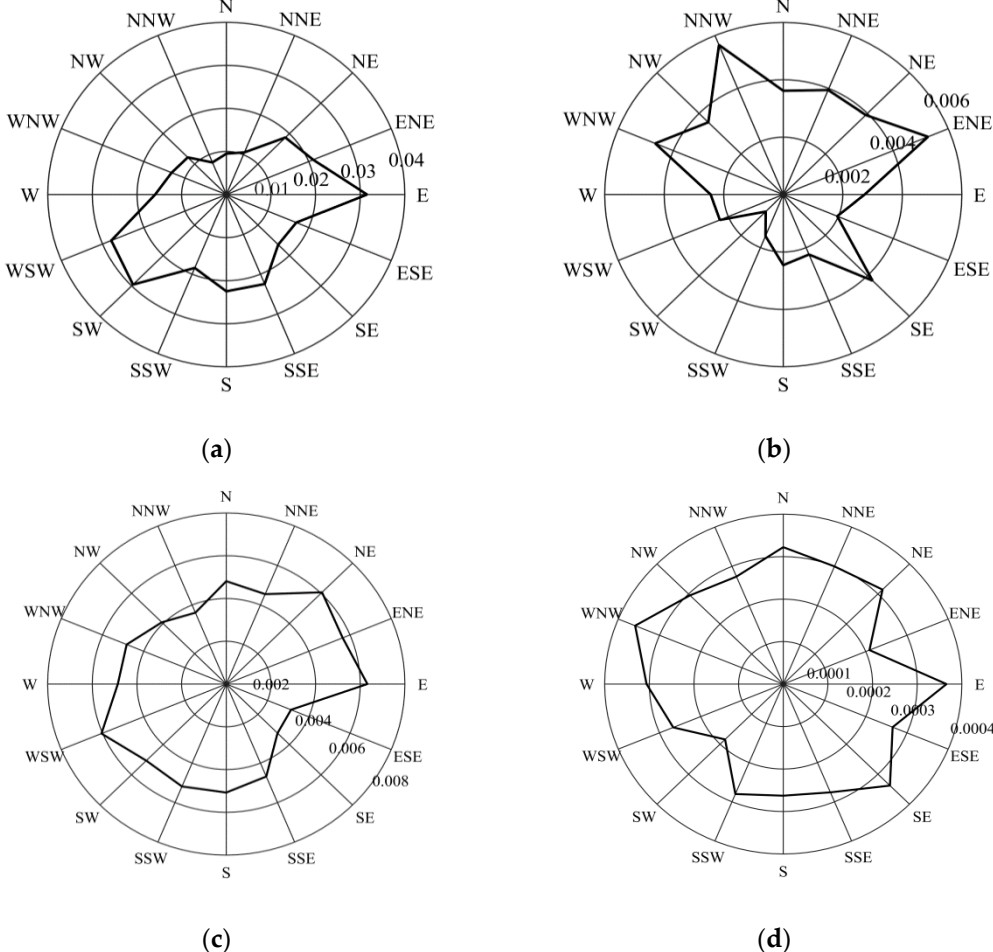

**Figure 7.** (**a**) Changes in $z_{0m}$ with different wind directions at Xiaotang from 5 January to 31 December 2010. (**b**) Same as (**a**) but for $z_{0h}$. (**c**) Same as (**a**) but for a magnitude of $10^{-3}$. (**d**) Same as (**b**) but for a magnitude of $10^{-4}$.

The diurnal variations in the means of $\ln(z_{0m})$, $\ln(z_{0h})$, and $kB^{-1}$ are shown in Figure 8. The range for $z_{0m}$ was $1.1 \times 10^{-3}$ to $8.4 \times 10^{-3}$ m, the range for $z_{0h}$ was $1.2 \times 10^{-4}$ to $6.1 \times 10^{-4}$ m, and $kB^{-1}$ varied from 0.9 to 3.9. The $\ln(z_{0m})$, $\ln(z_{0h})$, and $kB^{-1}$ terms were nearly constant during the daytime, but they fluctuated at sunrise and sunset (Figure 8). $z_{0m}$ depended on $u_*$ and $z/L$ (Figure 2h and Figure 10c). $\ln(z_{0m})$ was higher in July ($5.5 \times 10^{-3}$ m), but lower in November ($4.2 \times 10^{-3}$ m) (Figure 9a). Between December and June/July, the gradual increases in the wind velocity/surface-air temperature increased $H$ and $LE$ (Figure 9b–d). Blyth and Dolman [67] showed that the vegetation structure, meteorological conditions, and soil surface resistance were the main factors that affected $z_{0h}$, and thus $\ln(z_{0h})$ exhibited seasonal variations. We found that $z_{0h}$ was higher in April and September ($4.8 \times 10^{-3}$ m and $4 \times 10^{-3}$ m, respectively), but lower in January and February ($3.4 \times 10^{-3}$ m and $3.2 \times 10^{-3}$ m, respectively) (Figure 9a). The monthly variations in $\ln(z_{0h})$ correlated well with those in $LE$, but with a phase shift (Figure 9b).

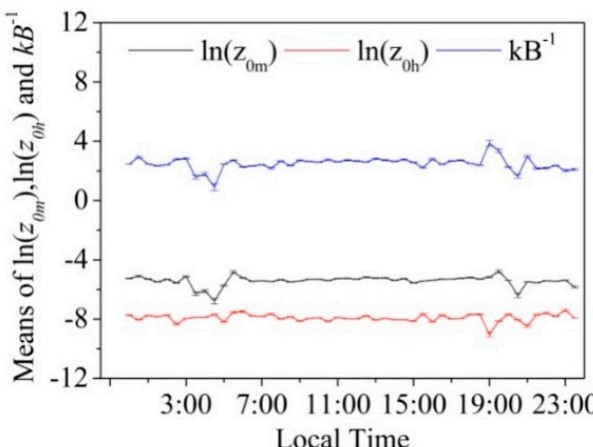

**Figure 8.** Diurnal variations in the mean values of $\ln(z_{0m})$, $\ln(z_{0h})$, and $kB^{-1}$ from 5 January to 31 December 2010.

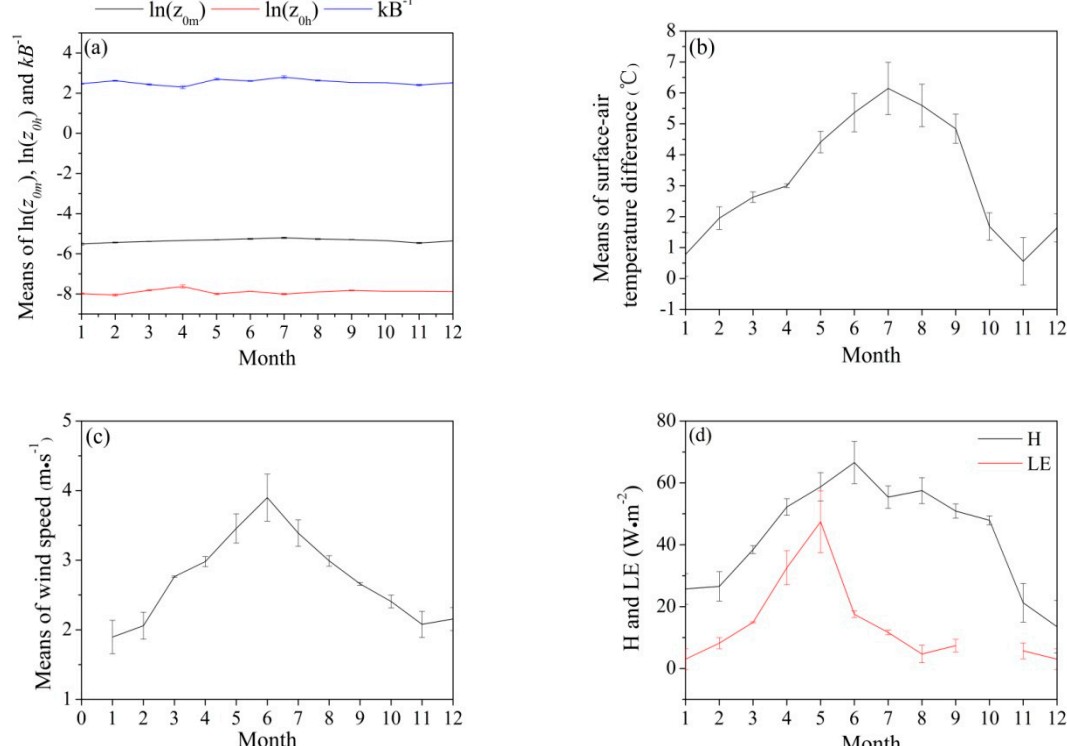

**Figure 9.** (**a**) Monthly variations in the mean values of $\ln(z_{0m})$, $\ln(z_{0h})$, and $kB^{-1}$ from 5 January to 31 December 2010. (**b**) Same as (**a**) but for the surface-air temperature. (**c**) Same as (**a**) but for the wind speed. (**d**) Same as (**a**) but for the sensible heat and latent heat fluxes.

In addition, the monthly variations in $z_{0m}$ and $z_{0h}$ were related to the surface conditions. For example, the station often experiences frequent dusty weather events during the spring and summer, and the number of dusty days in the spring (March to May) accounts for about 46% of the annual dusty days. Thus, $z_{0h}$ varied from $3.17 \times 10^{-4}$ m in February to $4.84 \times 10^{-4}$ m in April with a mean of $3.76 \times 10^{-4}$ m. In contrast to $\ln(z_{0h})$, the phase was the opposite for $kB^{-1}$. The $kB^{-1}$ term was nearly constant in all seasons and the monthly mean (based on January, February, March, April, May, June, July, August, September, November, and December) was 2.5. These results are similar to those obtained by Brutsaert [68] who found that the mean $kB^{-1}$ value was 2.3 for tall vegetation and it ranged between $-1.6$ to $-0.16$ for bare soil. Furthermore, our $kB^{-1}$ results for smooth surfaces were considerably different from that reported by Kondo [69] (i.e., $-1.1$). On the Qinghai–Tibetan

Plateau, the $kB^{-1}$ values were 2.5 and 2.36 in North PAM (Portable Automated Mesonet) and Anduo (both plateau meadows) [25], while in a semiarid region with degraded grassland and farmland, the $kB^{-1}$ values ranged among 7.1–9.2 and 8.6–11.0, respectively [70]. Verhoef et al. [23] showed that the $kB^{-1}$ values were 6.9, 8.1, 12.4, and −0.9 in HAPEX-Sahel (Hydrologic-Atmospheric Pilot Experiment in the Sahel), vineyard, sparse vegetation, and bare soil areas, respectively. This is because $z_{0m}$ increases with the vegetation height, so for a vegetated underlying surface, the vegetation height basically determines the magnitude of the roughness length. Therefore, with a tall vegetation underlying surface, the average value of $kB^{-1}$ was large. $z_{0m}$ was smaller in the desert than other areas, but the surface temperature difference and $H$ were larger in the desert than other areas. Thus, $kB^{-1}$ was different from that in other areas.

*3.6. Bulk Transfer Coefficients $C_d$ and $C_h$*

    According to Equations (15) and (16), the turbulent exchange is affected by $C_d$ and $C_h$, which are the key factors in turbulent flux parameterization schemes over different underlying surfaces. We estimated $C_d$ and $C_h$ according to Equation (15) (Eddy correlation method) and Equation (16) (M–O similarity theory), respectively. Figure 10 and Figure 13a,b shows the results obtained using Equation (15), and Figure 13c,d shows the results produced with Equation (16).

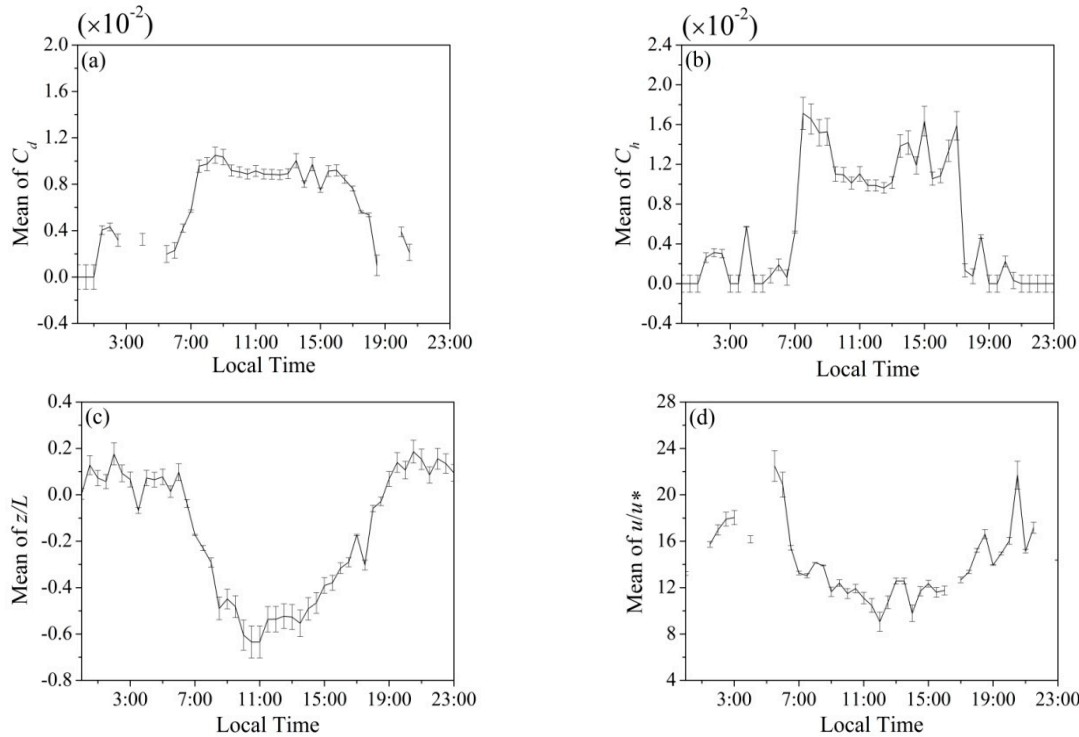

**Figure 10.** (**a**) Mean daily variation in $C_d$, with wind directions of NNE–ESE from 5 January to 31 December 2010. (**b**) Mean daily variations in $C_h$, with wind directions of NNE–ESE from 5 January to 31 December 2010. (**c**) Mean daily variations in $z/L$ with wind directions of NNE–ESE from 5 January to 31 December 2010. (**d**) Mean daily variations in $u/u_*$ with wind directions of NNE–ESE from 5 January to 31 December 2010.

    Figure 10a–d shows the variations in $C_d$ and $C_h$ according to their relationships with $z/L$ and $u/u_*$. Figures 8 and 10a,b shows that the changes in $z_{0m}$ and $z_{0h}$ were higher after sunrise and sunset, where they were consistent with the variations in $C_d$ and $C_h$. Figure 10a,b shows that the values of $C_d$ and $C_h$ were higher (lower) in the daytime (at night). According to the strong diurnal variations in the surface temperature, $z/L < 0$ after sunrise, but it was stable ($z/L > 0$) after sunset. Thus, $u/u_*$ was lower (higher) in the daytime (at night) (Figure 10d). As a result, $C_d$ and $C_h$ also exhibited diurnal variations,

where they responded to variations in the surface roughness length, atmospheric stratification and wind speed.

According to our observations, the mean values calculated over 24 h for $C_d$ and $C_h$ were $6.34 \times 10^{-3}$ and $5.96 \times 10^{-3}$, respectively. The mean values calculated during the daytime for $C_d$ and $C_h$ were $8.30 \times 10^{-3}$ and $1.11 \times 10^{-2}$, respectively. The mean values calculated during the nighttime for $C_d$ and $C_h$ were $2.44 \times 10^{-3}$ and $8.50 \times 10^{-4}$, respectively. Compared with previous studies, the mean $C_d$ and $C_h$ values were larger at the station than in the Gobi area (e.g., Dunhuang and HEIFE), the hinterland of the TD (e.g., Tazhong), and semiarid areas (e.g., Tongyu), but similar to the $C_d$ and $C_h$ values for the HEIFE desert [8,70–74]. These differences may be explained by the combined effects of the surface roughness and overlying atmospheric stratification. With the same underlying surface, the atmospheric stratification was more unstable and the bulk transfer coefficients were larger. $C_d$ was higher in near zero conditions with greater values of $z_{0m}$ [70].

Figure 11 shows the monthly variations in $C_d$ and $C_h$. Figure 11a,b shows that the highest values for $C_d$ and $C_h$ occurred in May and March ($0.99 \times 10^{-2}$ and $0.54 \times 10^{-2}$, respectively) whereas the lowest were in February and January ($0.30 \times 10^{-2}$ and $0.27 \times 10^{-2}$, respectively) because the atmospheric stratification was more unstable in the summer (the air temperature increased in the spring and summer). The order was $C_d > C_h$ (i.e., the momentum transfer was greater than the heat transfer) in all months except February (the standard deviation of $C_d$ was high in February) and the average ratio was 1.8.

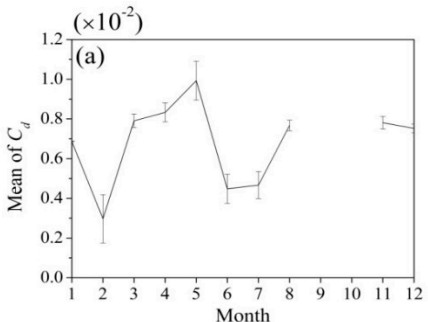 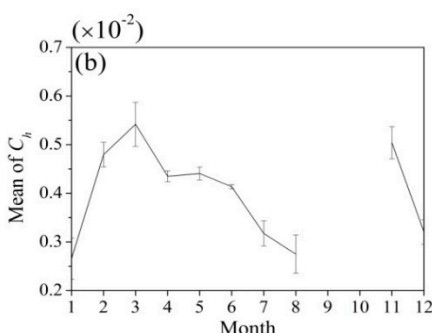

**Figure 11.** (**a**) Monthly variations in the mean values of $C_d$ under NNE–ESE wind direction from 5 January to 31 December 2010. (**b**) Monthly variations in the mean values of $C_h$ under NNE–ESE wind direction from 5 January to 31 December 2010.

The variation in $z_{0h}$ was low (from $5.11 \times 10^{-5}$ m to $7.90 \times 10^{-4}$ m), so both Equation (15) and Figure 9 indicate that the bulk transfer coefficients were mainly dependent on the wind speed and $z/L$, where the latter was dominated by the difference in temperature between the land surface and the atmosphere. According to our analysis, there was a good power function between the values of $C_d$ and $C_h$ and the wind speed (Figure 12). In general, Figure 12 suggests that $C_d$ and $C_h$ increased rapidly as the wind speed decreased under unstable conditions ($z/L < 0$) when the wind speed was less than 4.5 m s$^{-1}$.

By contrast, $C_d$ and $C_h$ exhibited nonlinear variations with the wind speed when the wind speed >4.5 m s$^{-1}$, where $C_d$ and $C_h$ approached constant values regardless of the wind speed while the atmosphere stratification tended to be neutral. These findings are similar to those reported by Feng et al. [70] who found that $C_d$ and $C_h$ decreased as the wind speed increased in unstable conditions and they were close to constant values as the wind speed increased with degraded grassland and cropland surfaces.

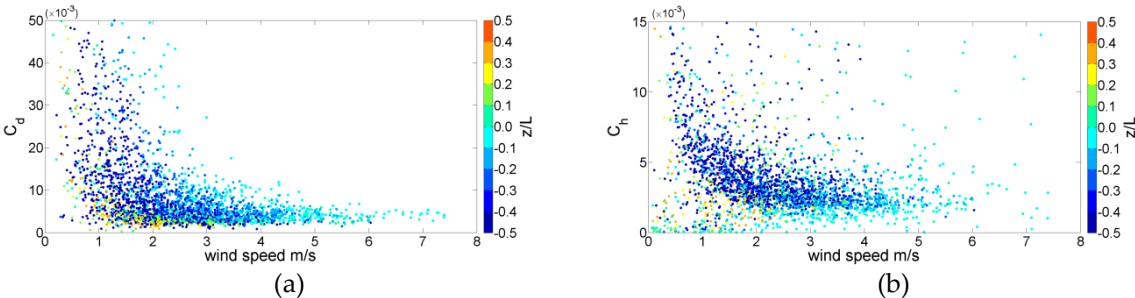

**Figure 12.** (**a**) Relationship between the $C_d$ and wind speed from 5 January to 31 December 2010.
(**b**) Relationship between the $C_h$ and wind speed from 5 January to 31 December 2010.

Figure 13a,b also shows the relationship estimated between $C_d$ and $C_h$ using the Eddy correlation method and the atmospheric stability in the NNE–ESE wind direction. For comparison, Figure 13c,d shows the results based on the estimates obtained using the M–O similarity function (Figure 3c is unrevised and Figure 3d is revised).

As shown in Figure 13, $C_d$ varied from $2 \times 10^{-3}$ to $10 \times 10^{-3}$ and $C_h$ from $0$–$9 \times 10^{-3}$ when $z/L$ ranged from $-3$ to $0.5$ at the station. $C_d$ and $C_h$ increased as $z/L$ decreased ($z/L \leq 0$), which is consistent with the results of previous studies in mid-latitudinal regions [4,70,75]. The variations in $C_d$ and $C_h$ versus the wind speed under different values of $z/L$ are shown in Figure 14. $C_d$ and $C_h$ decreased rapidly as the wind speed increased under weak wind conditions ($\leq 3.0$ m s$^{-1}$) and then remained almost constant as the wind speed increased further, where the minimum value was around $2$ m s$^{-1}$. By contrast, with the revised M–O similarity function, the difference was very small for $C_h$ but large for $C_d$.

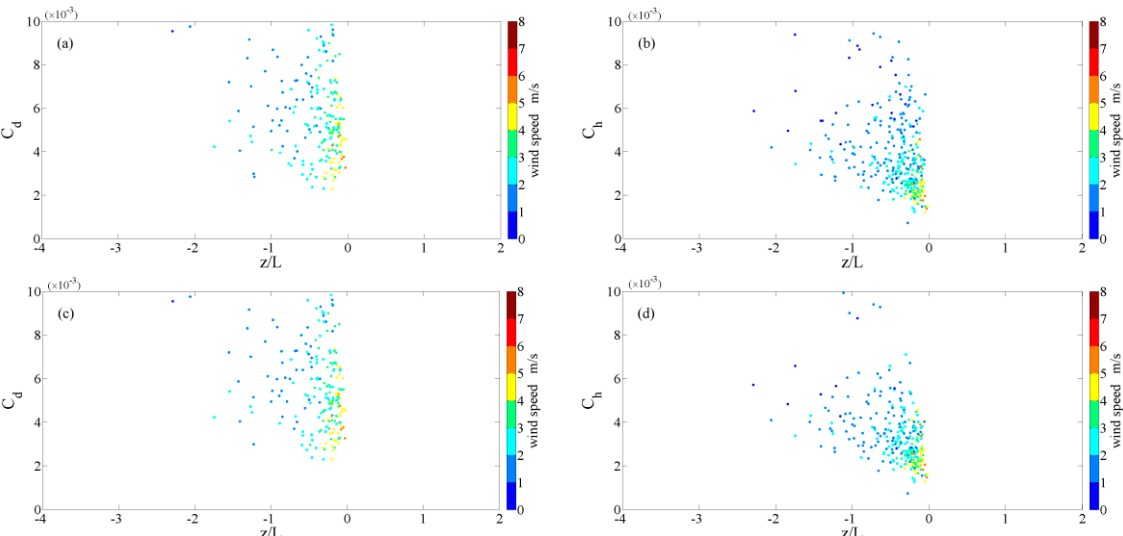

**Figure 13.** (**a**) Relationship between the bulk transfer coefficients and atmospheric stability with wind directions of NNE–ESE from 5 January to 31 December 2010 according to the Eddy correlation method. (**b**) Same as (**a**) but with wind directions of NNE–ESE. (**c**) Relationship between the bulk transfer coefficients and atmospheric stability with wind directions of NNE–ESE from 5 January to 31 December 2010 according to the unrevised M–O similarity function. (**d**) Same as (**c**) but revised.

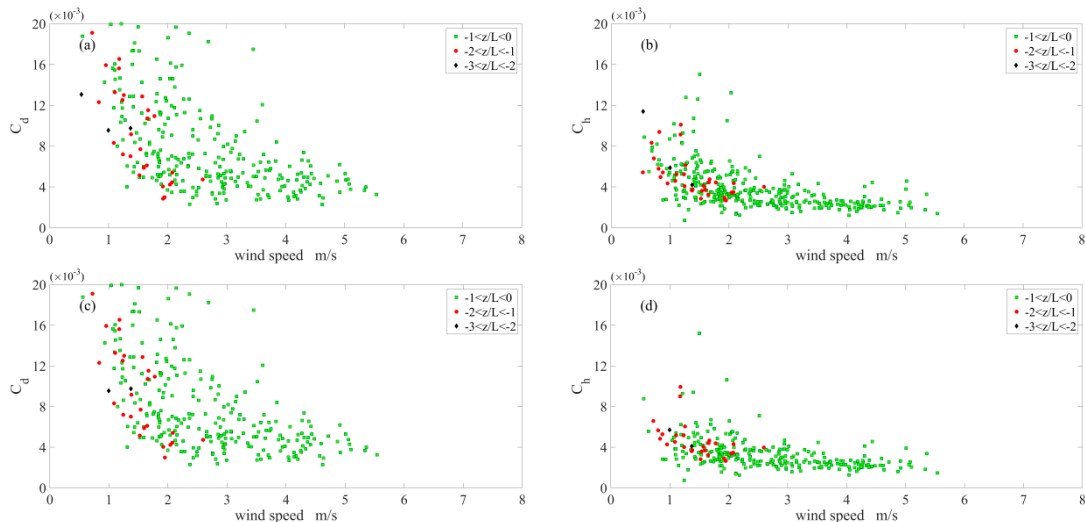

**Figure 14.** (**a**) Relationship between the bulk transfer coefficient $C_d$ and wind speed with wind directions of NNE–SES from 5 January to 31 December 2010. (**b**) Same as (**a**) but for $C_h$. (**c**) Same as (**a**) but according to the revised M–O similarity function. (**d**) Same as (**b**) but according to the revised M–O similarity function.

Table 6 shows the values of $C_d$ and $C_h$ calculated using the Eddy correlation method and the M–O similarity function (with NNE–ESE wind) method. In near natural conditions, the mean values of $C_d$ ($C_h$) were $20.72 \times 10^{-3}$, $4.30 \times 10^{-3}$, $4.23 \times 10^{-3}$, and $4.44 \times 10^{-3}$ ($6.62 \times 10^{-3}$, $1.70 \times 10^{-3}$, $2.74 \times 10^{-3}$, and $1.55 \times 10^{-3}$) with the Eddy correlation method (all wind directions), Eddy correlation method (with NNE–ESE wind), unrevised M–O similarity function (with NNE–ESE wind), and revised M–O similarity function (NNE–ESE), respectively. For the results derived based on all wind conditions, the mean values of $C_d$ ($C_h$) were $20.31 \times 10^{-3}$, $6.34 \times 10^{-3}$, $12.14 \times 10^{-3}$, and $7.94 \times 10^{-3}$ ($67.13 \times 10^{-3}$, $5.96 \times 10^{-3}$, $3.81 \times 10^{-3}$, and $3.33 \times 10^{-3}$), respectively. The values of $C_d$ ($C_h$) at the station calculated using the Eddy correlation method (all wind direction) were larger than those calculated with the other three methods, i.e., Eddy correlation method (with NNE–ESE wind direction), and unrevised and revised M–O similarity function (with NNE–ESE wind direction). The values of $C_d$ ($C_h$) at the station calculated with the unrevised M–O similarity function (NNE–ESE) were larger than those calculated with the revised M–O similarity function (NNE–ESE).

**Table 6.** Values of $C_d$ and $C_h$ calculated using different methods.

| Methods Bulk Transfer Coefficients ($\times 10^{-3}$) | Eddy Correlation Method | | | | M–O Similarity Function (NNE–ESE Wind) | | | |
|---|---|---|---|---|---|---|---|---|
| | All Wind Directions | | NNE–ESE Wind | | Unrevised | | Revised | |
| | $C_d$ | $C_h$ | $C_d$ | $C_h$ | $C_d$ | $C_h$ | $C_d$ | $C_h$ |
| Near natural conditions | 20.72 | 6.62 | 4.30 | 1.70 | 4.23 | 2.74 | 4.44 | 1.55 |
| Average | 20.31 | 7.13 | 6.34 | 5.96 | 12.14 | 3.81 | 7.94 | 3.33 |

We found that the estimates of $C_d$ and $C_h$ varied with different wind directions. Under the NNE–ESE wind direction, the values of $C_d$ and $C_h$ decreased (increased) as the wind speed increased under stable conditions (unstable conditions). For these wind directions, the values of $C_d$ or $C_h$ had the same magnitude using the Eddy correlation method and the revised M–O similarity function method. These results were related mainly to the flux source at the station, i.e., the maximum flux source area in 90% of cases was in the E–SE direction from the 3-m tower. The SW direction flux was affected by the 3-m tower. By contrast, the areas to the SE and NW from the 3-m tower comprised a flat ancient river bed, and those in the NE and SW directions from the 3-m tower were high and rolling dunes.

## 4. Discussion

Compared with other underlying surfaces around the world, desert is quite distinctive in the physical process of the underlying surface, especially the heating process between the land surface and the atmosphere. Desert has higher $\alpha$, lower $\varepsilon$, $z_{0m}$ and $z_{0h}$, compared to humid areas. Due to the remarkable land heating effect on the atmosphere in the desert, the mixing layer develops much deeper than other areas, and the Bowen ratio shows significant variations [76]). Sandstorms occur frequently in desert, resulting in obvious changes in the surface absorption of solar radiation and the proportion of sensible heat flux, latent heat flux and soil heat flux by the direct and indirect radiation effects of dust aerosols. Also, parameters ($\alpha$, $\varepsilon$, $z_{0m}$, $z_{0h}$, $kB^{-1}$, $C_d$, and $C_h$) were influenced in sandstorms weather. Consequently, it will influence the land-atmosphere interaction in the desert greatly. In addition, because of the extremely low precipitation and strong evaporation in desert, there is few water vapor in the soil as well as atmosphere. Therefore, the energy transfer of the desert is dominated by sensible heat flux, while the latent heat flux is very small, which distinguishes the desert from other areas. What is more, land-atmosphere interaction in desert plays an important role in global and regional energy balance and climate change [77].

## 5. Conclusions and Further Work

The overland-surface processes in the desert region of northwest China play important roles in the regional weather and climates. $\alpha$, $\varepsilon$, $z_{0m}$, $z_{0h}$, $C_d$, and $C_h$ are key parameters for the land–air interactions. In this study, these parameters were investigated based on observations on the northern marginal zone of TD and compared with the values estimated from the observations in the central area of TD. The main findings are summarized as follows.

(1) In the northern marginal zone of the TD, $\alpha$ and $\varepsilon$ were 0.27 and 0.91, respectively, which are consistent with the values obtained based on observations in the hinterland of the TD as well as being similar to the dry parts of the Great Basin desert in North American. Also, the retrieved $\alpha$ and $\varepsilon$ from Landsat 8 data were close to the observed values at both the northern marginal zone and hinterland of the TD. The satellite-derived values of $\alpha$ and $\varepsilon$ from remote sensing products are highly reliable for use in the Taklimakan Desert. The values of $\alpha$ varied from sunny to cloudy days, and were 0.31 and 0.30, respectively. Also, the relationship between $\alpha$ and the solar elevation angle ($h$) during sunny periods and cloudy periods were different, which were $\alpha = 0.2586 + 0.2415e^{-h/8.852}$ and $\alpha = 0.4424h^{-0.1404}$, respectively. The values of $\varepsilon$ were sensitive to the observed value of $H$ using the iterative method, that $\varepsilon$ decreased (increased) as $H_{obs}$ increased (decreased).

(2) In the marginal zone of the TD, $z_{0m}$ was dependent on the wind direction. $z_{0m}$ clearly changed with the wind direction, which could be affected by the spatial inhomogeneity (The distributions of the surrounding sand dunes at the station are not uniform, there were many undulating sand dunes in the prevailing wind and opposite to the prevailing wind). These are similar to the results obtained in vegetated areas where $z_{0m}$ was highly dependent on the seasonal variations in vegetation conditions. In general, in the NE–E and SSE–WSW directions, there were many undulating sand dunes, which were consistent with the directions of the peak $z_{0m}$ value, which $z_{0m}$ was $5.6 \times 10^{-3}$ to $6.3 \times 10^{-3}$ m in the NE–E wind direction and $5.1 \times 10^{-3}$ to $6.1 \times 10^{-3}$ m in the S–WSW wind direction. $z_{0h}$ changed with the wind direction where the magnitudes of the data were $10^{-3}$ and $10^{-4}$ for $z_{0m}$ and $z_{0h}$, respectively. The optimal values for $z_{0m}$ and $z_{0h}$ were $5.858 \times 10^{-3}$ m and $1.965 \times 10^{-4}$ m according to the normally distributed histogram, and the magnitudes of $z_{0m}$ and $z_{0h}$ at the station were consistent with the results reported by Stull [63].

(3) $z_{0m}$ and $z_{0h}$ varied seasonally. $Z_{0m}$ was higher in July ($5.5 \times 10^{-3}$ m) and lower in November ($4.2 \times 10^{-3}$ m); $z_{0h}$ was higher in April and September ($4.8 \times 10^{-3}$ m and $4 \times 10^{-3}$ m, respectively) and lower in January and February ($3.4 \times 10^{-3}$ m and $3.2 \times 10^{-3}$ m, respectively). The monthly $z_{0m}$ varied between $3.2 \times 10^{-3}$ (February) and $4.8 \times 10^{-3}$ m (April), and the average annual value was $4.78 \times 10^{-3}$ m. The monthly $z_{0h}$ varied between $3.17 \times 10^{-4}$ (February) and $4.84 \times 10^{-4}$ m (April), and the average annual value was $3.76 \times 10^{-4}$ m. The values of $\ln(z_{0m})$, $\ln(z_{0h})$, and $kB^{-1}$ exhibited obvious

fluctuations at sunrise and sunset. The excess resistance to heat transfer, $kB^{-1}$, varied inversely with $\ln(z_{0h})$. The daily mean $kB^{-1}$ was between 0.9 and 3.9, and the average annual value was 2.5. These results were different from those obtained in the vegetated areas with bare soil and smooth surfaces such as the HEIFE Gobi desert, but they were similar to those determined for the Qinghai–Tibetan Plateau and HAPEX-Sahel.

(4) Due to the high $H$ value in this area, $C_d$ and $C_h$ exhibited large diurnal variations in the marginal zone of TD. The fluctuations in $C_d$ and $C_h$ were large before sunrise and after sunset, and they were consistent with $z_{0m}$. The order was $C_d > C_h$ in all months except February. The daily mean values of $C_d$ and $C_h$ were $6.34 \times 10^{-3}$ and $5.96 \times 10^{-3}$, respectively, and their neutral values were $4.30 \times 10^{-3}$ and $1.70 \times 10^{-3}$. The highest and lowest values for $C_d$ were $0.99 \times 10^{-2}$ (May) and $0.54 \times 10^{-2}$ (March), respectively, while the highest and lowest values for $C_h$ were $0.30 \times 10^{-2}$ (February) and $0.27 \times 10^{-2}$ (January).

Our results also suggested that the parameters derived from remote sensing observations were good quality compared with the in situ observations. Therefore, remote sensing provides a better method for capturing the characteristics of the surface parameters and land-surface processes in a vast desert region where in situ observations are very sparse and difficult to maintain. In future research, we will use our observations to evaluate whether the uncertainty of parameters derived from observations can affect the capacity of land-surface models for modeling land–air interactions in a desert climate. We will also assess how much the uncertainty in surface parameter estimates can affect simulations of the desert weather and climate in this region.

**Author Contributions:** L.J. conceived and designed the analysis; L.J., Z.L., Q.H., Y.L., Y.X. analyzed the data; L.J., Z.L. processed the data; A.M., X.L., W.H., J.Z., C.Z. observed and collected the data in the Taklimakan Desert; L.J. wrote the paper.

**Funding:** This research was funded by [Basic Business Expenses] [IDM201505)], [Meteorology Public Welfare Industry Research Special Project] [GYHY(QX)201506001-14], [the National Natural Science Foundation of China] [41605008], [China Postdoctoral Science Foundation] [2016M592915XB] and [Flexible Talents Introducing Project of Xinjiang (2016)].

**Conflicts of Interest:** The authors declare no conflict of interest.

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
