# Peer review of "Observed Key Surface Parameters for Characterizing Land–Atmospheric Interactions in the Northern Marginal Zone of the Taklimakan Desert, China"

_atmosphere, doi:10.3390/atmos9120458_

Round 1

Reviewer 1 Report

This is a nice piece of work. It is very well written and detailed in its contents. I'd appreciate more discussion about the comparison/agreement between satellite and observation results. I'd also like to see in the Discussion a section referred to land - atmosphere interactions in deserts, which will increase the global interest of the manuscript. Land - atmosphere interaction in desert is an overlooked ground in the filed and is worth to discuss its implications. In my opinion, no further revisions are needed for this work. 

Reviewer 2 Report

Review of the article: Observed Key Surface Parameters for Characterizing Land–Atmospheric Interactions in the Northern Marginal Zone of the Taklimakan Desert, China by Jin et al.

General comments: The article provides a nice summary of the estimates and variability of land - atmosphere interaction parameters. The results are interesting, expected and relevant and I thank the authors for presenting in a clear and concise fashion. Just some minor suggestions:

Specific comments:

L56: you mean neutral and thermally stratified conditions, respectively.

L59: more literature suggestions:

Banerjee, T., De Roo, F. and Mauder, M., 2017. Explaining the convector effect in canopy turbulence by means of large-eddy simulation. Hydrology and Earth System Sciences (Online), 21.

Mauder, Matthias, et al. "Evaluation of energy balance closure adjustment methods by independent evapotranspiration estimates from lysimeters and hydrological simulations." Hydrological Processes 32.1 (2018): 39-50.

L65-68: not sure the I understand the cause-effect statement here. If we do not understand all the feedbacks involved in land atmosphere interaction (which is true), we can focus on a particular subset of the problem. Why would we focus on an area in China?

L69: How do soil organic carbon and soil moisture influence albedo? what is the mechanism? With more carbon, shouldn’t the surface be darker, reducing albedo? Please explain.

L72: What do You mean by the surface energy budget has variations? Is it the energy balance residual or the individual terms of the budget?

L93: I would avoid ‘disturbance’ here. You can say distortions in the measurements.

L232: please comment on the formulation for r_h.

Figure 14: Please use different colors. Hard to observe.
